# The Functioning of Hospice in the Perception of Family Members of Cancer Surgery and Hospice Patients

**DOI:** 10.3390/ijerph20075334

**Published:** 2023-03-30

**Authors:** Paulina Aniśko-Trembecka, Magda Popławska, Elżbieta Krajewska-Kułak, Irena Mickiewicz, Wojciech Kułak

**Affiliations:** 1Doctoral Studies, Medical University of Białystok, 15-089 Bialystok, Poland; 2Student Research Group, Department of Integrated Medical Care, Medical University of Bialystok, 15-089 Białystok, Poland; 3Department of Integrated Medical Care, Medical University of Bialystok, 15-089 Bialystok, Poland; 4Independent Public Palliative Care Team for Them, John Paul II in Suwałki, 16-402 Suwałki, Poland; 5Department of Pediatric Rehabilitation, Center of Early Support for Handicapped Children “Give a Chance”, Medical University of Białystok, 15-274 Bialystok, Poland

**Keywords:** perception, hospice, families

## Abstract

Background: Palliative care in Poland is for all dying people and their families to have timely access to quality care services. The study aimed to assess the perception of the role of hospice care by families of patients treated in oncological surgery departments and hospices. Methods: The study included 211 family members of cancer patients, comprising 108 family members of cancer surgery patients (Group I) and 103 hospice patients (Group II). The study used a diagnostic survey method with a proprietary questionnaire. Results: 74.9% of people in Group I and 84.6% in Group II experienced positive associations with hospice care. 86% of respondents from Group I believed that hospice is a place where patients can die with dignity, while 68.3% of those from Group II believed it is where patients receive professional care. 56.7% from Group I and 65.4% from Group II did not feel anxious about hospice care. According to 68.6% of people in Group I, informing the patient that he or she is in hospice as well as about his or her disease should depend on the patient’s condition. In the opinion of 75% of Group II, the patient should always be informed. In Group I (68.3%) and Group II (91.5%), the dominant opinion was that the family should take part in the care and treatment of the patient. 78.4% of respondents in Group I and 96.4% in Group II recommend hospice to other families. Conclusions: Most families of cancer patients from both the oncological surgery departments (Group I) and hospice (Group II) had positive first associations with hospice care. However, families from Group II had more critical remarks on hospice functioning.

## 1. Introduction

Palliative and hospice care are designed to relieve pain and distressing symptoms of a disease and maintain the patient’s quality of life at the highest possible level [1,2]. In Poland, the National Health Fund finances several types of such care: outpatient palliative medicine clinics, stationary hospice or palliative medicine departments in a hospital, and home hospice [3]. Palliative and hospice care covers mainly people with cancer, AIDS, consequences of diseases of the central nervous system, some types of respiratory failure, cardiomyopathy, chronic wounds, and ulcers from bedsores. Cancer is Poland’s leading principal diagnosis of hospice patients (90%).

A diagnosis of cancer changes the entire family’s life and the patient’s immediate environment [4]. People in the patient’s immediate vicinity express strong emotions, worries, and fears. Cancer never only affects a sick person. It affects the everyday life of the whole family and forces a change of roles in the family, along with the rhythm of the day and daily habits. In many cases, it also changes the economic status of the family. Thus, family members become so-called “second-line patients” [5].

Psycho-oncologists have distinguished four characteristic attitudes that families have towards neoplastic disease, and families may present alternating attitudes or periodically exhibit features typical of each of them [6]:Active cooperation—active, close-attitude cooperation of the whole family with the patient and doctors; family members are open to communication and honesty; they take the doctor’s advice; many of them frequently come up with their initiatives.Task-related—limited to passive esteem; family members show no particular commitment or initiative.The critical—negative perception of the treatment process and its effects; criticizing medical personnel; attempting treatment on their own, which requires a firm response from doctors, as it may harm the patient.Expressed in total disengagement—family members assume that they are obliged to treat and care for the sick on their own, disregarding the responsibility of doctors and medical staff.

Patients with advanced cancer often experience pain, dyspnea, and distressing respiratory symptom, and use intensive, hospital-based services near death [7]. Hospice offers an alternative, patient-centered model of care focused on relieving suffering, and often delivers services within the home environment [8].

It has been accepted in Polish culture that parents help their children enter adult life, who in turn later serve their parents in their old age. However, it is not uncommon for people to face situations where the only solution is to commit a loved one to a care and treatment institution or a hospice. These types of decisions are never easy.

There are only a few studies on the perception hospice care by families in Poland [9].

The study aimed to compare the perception of the role of hospice by families of patients treated in oncological surgery departments and hospices.

## 2. Materials and Methods

This is a quantitative study. In the group of families in oncological surgery (Group I), 130 questionnaires were distributed, and 108 questionnaires were returned—50.8% from women and 49.2% from men. In the group consisting of families of patients in hospice (Group II), 150 questionnaires were given, and 104 were returned, 67.3% of responses from women and 32.7% from men.

### 2.1. Ethics

The research was carried out after obtaining the approval of the Bioethics Committee no. APK.002.175.2020 of the Medical University of Bialystok, Poland.

All methods were performed in accordance with the Declaration of Helsinki.

### 2.2. Data Collection

The study used the original questionnaire, which contained questions about the respondents’ opinions on pain, suffering at the end of life, feelings during contact with the dying, the patient’s struggle for life when facing the inevitability of death, euthanasia, the perception of hospice, hospice requirements, the need to inform the patient about their illness and the fact that they are in hospice, the hospice staff, the family’s participation in the care/treatment of the patient, the patient’s feeling of loneliness, the expected support from hospice, the desired religious practices in hospice, recommending hospice to other families as a form of care, public attention to palliative care, problems in patient care regarding those facing death, and opportunities to improve patient care at the end of life.

### 2.3. The Statistical Analysis

The statistical analysis was conducted using Statistica 13 P.L. Results are presented as mean values ± S.D. Chi^2^ test with Yates’ correction and Fisher exact test were used to compare the percentage answers of hospice patients’ families with patients’ families on the oncological surgery ward. The critical level for all tests of significance was *p* < 0.05.

## 3. Results

According to 64.8% of respondents in Group I and 51% in Group II, pain and suffering at the end of life is primarily a physical experience. Other indications are presented in Table 1.

The respondents from Group I (48.2%) and Group II (50.0%) declared that they most often felt anxiety during contact with the dying. 59.4% of family members from Group I and 13.8% from Group II did not avoid contact with the dying person. Other indications are presented in Table 2.

The majority of respondents from Group I (72.1%) reported that it was absolutely necessary to fight for the patient’s life in the event of the inevitability of death. Only 37.2% of families from Group II said the same. Other indications are presented in Table 3.

Most people (75.2% from Group I and 45.2% from Group II) were against euthanasia. In Group I (35.8%), the respondents did not consider euthanasia methods to be applicable. Group II (46.2%) considered the exclusion of life-support devices as an acceptable method. Other indications are presented in Table 4.

It was shown that 74.9% of Group I and 84.6% of Group II family members had positive initial associations with hospice care. Most respondents thought that hospice did not drive anxiety (56.7% from Group I and 65.4% from Group II). In Group I, the prevailing opinion was that hospice was a place where patients could die with dignity (86%). Most respondents reported that patients should be in hospice because the hospice staff were greatly supportive, showing much kindness and courtesy (92.5% from Group I and 100% from Group II). The remaining results are presented in Table 5.

According to 47.5% of family members in Group I and 67.5% in Group II, the waiting time for admission into hospice should be less than 8 days. The remaining results are presented in Table 6.

The majority of respondents from Group I (68.6%) agreed that informing the patient that he/she is in hospice is important and that illness should depend on his/her physical, mental, and spiritual conditions. 75% of those in Group II shared similar opinions. The remaining results are presented in Table 7.

The majority of respondents from Group I (98.2%) claimed that any doctor (98.2%) and nurse (97.5%) could work in hospice care. According to respondents from Group II, only doctors specializing in palliative medicine (78.8%) and nurses specializing in or taking a course in palliative medicine (62.5%) should work in hospice care. The remaining results are presented in Table 8.

According to Group I (68.3%) respondents and Group II respondents (91.5%), families should take part in the care/treatment of the sick person. The remaining results are presented in Table 9.

Respondents from Group I (54.7%) and Group II (62.4%) reported that hospice patients do not feel lonely. They agreed that support should be provided primarily to the sick (98.7% from Group I and 85.6% from Group II), and it should be provided mainly by a clergyman. The remaining results are presented in Table 10.

The most desirable religious practice in hospice was Catholic mass (86.7% from Group I and 87.9% from Group II). The remaining results are presented in Table 11.

According to 87.3% of family members of patients from Group I and 59.7% from Group II, society does not pay much attention to palliative care. The remaining results are presented in Table 12.

The subjects from Group I saw problems with patient care at the end of a patient’s life, mainly in the increasing number of patients with chronic diseases (96.7%). Moreover, respondents from Group II reported that financial expenditures on hospice were too low (50%). Details are presented in Table 13.

In the opinion of the greatest number of respondents from Group I, improving the quality of medical care in hospice would result in an increase in the number of medical personnel (87.4%) and an increase in the number of beds (82.4%), and would be seen as showing more kindness and support for patients and families (92.8%) in (Group II). Details are presented in Table 14.

Most respondents from both groups recommended hospice as the place of death.

## 4. Discussion

Most families from both studied groups had positive first associations with hospice. The diversity of opinions of the surveyed families on the functions of hospice was noted. For example, families of hospice patients (Group II) had more remarks on the scope of information, the role and composition of the staff, the participation of families in the care/treatment of the patient, the patient’s feeling of loneliness, the expected support, and desired religious practices in hospice.

In the present study, the most significant number of people (75.2% from Group I and 45.2% from Group II) were against euthanasia. In Group I, 35.8% of the respondents did not consider any methods of euthanasia to be applicable, and 46.2% of respondents in Group II considered turning off life-support apparatus as an acceptable method.

The word “hospice” is often understood to be a place for patients and their families to spend the last days of the patient’s life, simply “waiting for death.”

In the study by Łukaszuk et al. [10], 48% of respondents had negative thoughts associated with hospice. However, 47% of the respondents thought positively about it. Different results were obtained in the present study: 74.9% of family members of patients from oncological surgery departments and 84.6% of families of hospice patients had positive first associations with hospice, and most often, they did not feel fear when thinking about hospice.

Hospices should also provide access to social services until the patient’s death. Death is treated as a normal process and is not hastened or delayed. Hospice also offers hope of relieving pain and other troublesome symptoms through professional care. Many studies have shown the effects of palliative care in reducing disease symptoms and improving quality of life. However, their interpretation is difficult to decipher due to methodological differences regarding the time of the study’s conduct and the small number of patients [11,12,13,14,15,16]. In the current study, the group of families of patients in oncological surgery wards had the prevailing opinion that a hospice is a place where patients can die with dignity, and the group of families of patients in hospice stated that it is a place where patients receive professional care.

Research conducted by CBOS in 2009 [17] showed that Poles almost universally speak of palliative care with appreciation. The operation of home hospices was supported by 94% of respondents and 96% of inpatient hospices. However, despite the widespread social support for hospice activities, most respondents believed that inpatient hospice is where dying people should go only in exceptional circumstances, e.g., when the family cannot provide them with specialist care. The respondents from the current study believed that hospice was the only appropriate place for terminally ill people. At the same time, the surveyed members of both families expressed the conviction that hospice was a place only for people with cancer, and the families of surgical patients more often claimed that this place was only for patients just before death (53.7% vs. 6.7%)

Leppert et al. [18] compared students’ and doctors’ difficulties in communicating unfavorable news about a disease and prognosis. The percentage of respondents ready to provide complete information was only 28% among students and 24% among doctors. Most patients (85%) from a study by Mess et al. [19] claimed that a person should always be informed if they have an incurable disease. According to Coughlan [20], most patients receiving chemotherapy for cancer knew their diagnosis in Ireland. In contrast, only 32% of patients in Spain received information about their current health status [21]. In a study by Chua et al. [22], almost all patients wanted information about the disease, tests, research, treatment, side effects, psychosocial support, and financial issues. The results of other studies also confirm that cancer patients have many information needs, and the lack of such information leads to increased anxiety [23,24]. In the opinion of most of the surveyed family members of patients in the oncological surgery ward, informing the patient that he or she is in hospice and about his or her disease should depend on the patient’s physical, mental, and spiritual conditions.

Spiritual care is integral to providing quality end-of-life care [25,26]. However, patients often report that this aspect of care is lacking. For example, in a study by Bejda et al. [27], the presence of chaplains or the holy Mass in hospice did not arouse any dissatisfaction. Similarly, in the present study, the respondents from both groups considered the mass and the rosary prayer as a desirable religious practice in hospice.

Patients and family caregivers valued the personal qualities of staff, their experiences and specialized knowledge and skills, and the development of a close rapport amongst staff, patients, and their families [28,29]. However, common reasons for patients’ dissatisfaction with hospice care include too-rare contact with medical professionals involved in their care, poor communication, insufficient care equipment, and a lack of emotional support from the family.

In the opinion of the most significant number of surveyed families of patients in oncological surgery wards, the improvement of the quality of medical care in hospices would result from an increase in the number of medical personnel and an increase in the number of beds, along with an increased demonstration of kindness and support for patients and their families.

It is worth noting that the family’s participation in palliative care is usually very high [30]. In addition, family members play an important role in providing emotional support to patients [31,32]. In the current study, most of the respondents from both groups were convinced that the family should take part in the care/treatment of the patient.

It should be noted that families can often feel shame that they have “given” their loved one to hospice. Therefore, it is necessary to continually increase society’s awareness about hospice care, emphasizing that every family’s situation is different, and each patient’s case is considered individually in hospice. Unfortunately, in the public consciousness, palliative treatment is still associated with the terminal stage of neoplastic disease. It is often associated with an unrealistic perception of the possibilities and effectiveness of oncological treatment. It is, therefore, necessary to promote the idea that a patient in a hospice is entrusted with professional care, and all efforts made there are aimed at improving the patient’s quality of life.

## 5. Study Limitations

The limitations of the current research include the insufficient number of members of the studied families and the assessment of families from only the oncological surgery departments and not from the chemo or radiotherapy departments. In the future, it would also be worth checking what the tested opinions are influenced by, e.g., the respondents’ life satisfaction and previous experiences with a chronic disease in the family.

## 6. Conclusions

Most families from both studied groups had positive first associations with hospice; they did not fear it when they thought about it and would recommend it to other families as a form of care for the sick.

The diversity of the opinions of the surveyed families on the functions of hospice was noted. For example, families of hospice patients (Group II) had more remarks on the scope of information, the role and composition of the staff, the participation of families in the care/treatment of the patient, the patient’s feeling of loneliness, the expected support, and desired religious practices in hospice.

Families of patients from oncological surgery departments were more convinced than the families of patients in hospice that society does not pay much attention to palliative care.

## Figures and Tables

**Table 1 ijerph-20-05334-t001:** Respondents’ opinions about pain and suffering at the end of life *.

	Patients’ Families with		*p* Value
	Oncological Surgery Ward (Group I) N = 108	Hospice (Group II) N = 104	
Sensation primarily physical	64.8%	51.0%	NS
An experience preventing normal functioning	56.2%	46.2%	NS
Spiritual experience	32.9%	31.7%	NS
Mental experience	32.8%	42.3%	NS
Element of human existence	27.8%	16.3%	NS
Punishment for sins	1.2%	5.8%	NS
Evil	0.8%	1.9%	NS
Grace	0%	5.8%	0.046
Difficult to say	0%	6.7%	0.023

* Possibility of multiple-choice answers.

**Table 2 ijerph-20-05334-t002:** The respondents’ opinions about their feelings during contact with the dying.

	Patients’ Families with		*p* Value
	Oncological Surgery Ward (Group I) N = 108	Hospice (Group II) N = 104	
Anxiety			
Presence	48.2%	50.0%	NS
Lack	26.1%	36.2%	NS
Difficult to say	25.7%	13.8%	NS
Avoiding contact			
No	59.4%	13.8%	<0.001
Yes	28.9%	72.2%	<0.001

**Table 3 ijerph-20-05334-t003:** The respondent’s opinions about the struggle for the patient’s life in the event of the inevitability of death.

	Patients’ Families with		*p* Value
	Oncological Surgery Ward (Group I) N = 108	Hospice (Group II) N = 104	
Absolutely should fight	72.1%	37.2%	0.019
Death is a natural phenomenon that must be considered and fought as far as possible. But it must be accepted if it is inevitably approaching.	82.4%	66.0%	NS

**Table 4 ijerph-20-05334-t004:** The respondents’ opinions about euthanasia.

	Patients’ Families with		*p* Value
	Oncological Surgery Ward (Group I) N = 108	Hospice (Group II) N = 104	
Opposed to euthanasia	75.2%	45.1%	0.035
Pro euthanasia	14.4%	14.2%	NS
For euthanasia under certain conditions (the disappearance of all functions of the body except the heart. Expressing such will by the patient. Death of the brain stem. Very old age. Incurable disease. Cases when medicine has used its capabilities, and there is no chance of improvement.)	5.3%	9.3%	NS
Difficult to say	5.1%	31.4%	<0.001
Acceptable forms of euthanasia			
Shutdown of supporting apparatus	25.8%	46.2%	0.046
Abandonment of the resuscitation action	25.6%	21.1%	NS
Administration of a drug in a lethal dose	12.8%	9.6%	NS
Not considering any of these methods as applicable	35.8%	23.1%	NS
Legal recognition of euthanasia			
Yes	16.7%	19.0%	NS
No	53.1%	43.0%	NS
Difficult to say	30.2%	38.0%	NS

**Table 5 ijerph-20-05334-t005:** The respondents’ opinions about hospice.

	Patients’ Families with		*p* Value
	Oncological Surgery Ward (Group I) N = 108	Hospice (Group II) N = 104	
First associations with hospice			
Positive	74.9%	84.6%	NS
Negative	9.1%	6.7%	NS
Hard to say	16%	8.7%	NS
Thinking about a hospice			
Feeling anxious	19.3%	34.7%	
Not feeling anxious	56.7%	65.4%	NS
Hard to say	24%	0.1%	<0.001
Hospice is… *			
comprehensive care for a terminally ill patient.	66.7%	57.7%	NS
the right place for terminally ill people because the family is not always able to provide proper care.	61.8%	51.0%	NS
the only right place for terminally ill people.	50.4%	18.3%	0.002
an institution providing inpatient palliative care.	33.3%	47.1%	NS
a place where the patient has a chance to fight for life and live with dignity.	31.9%	22.1%	NS
a place where patients in hospice receive professional care.	79.1%	68.3%	NS
a place where the sick can die with dignity.	86%	45.2%	0.012
a place where the sick hate.	10.2%	22.1%	NS
a place where patients find relief and peace.	56.1%	28.8%	0.013
a place where patients suffer and die.	12.5%	3.8%	NS
a place for people suffering from an incurable disease. regardless of the diagnosis.	65.6%	52.9%	NS
a place for people only with cancer.	59.2%	55.8%	NS
a place only for patients just before death.	53.7%	6.7%	<0.001
hard to say.	6.7%	0.0%	0.029
Information provided in hospice is something that… *			
should be comprehensive.	89.8%	78.6%	NS
the doctor should be happy to give them.	97.5%	85.7%	NS
the nurse should be happy to give them.	96.8%	100.0%	NS
the family should obtain full information about the patient’s prognosis,	76.4%	87.8%	NS
the family should not obtain complete information about the patient’s prognosis.	11.1%	5.5%	NS
the patient and family should be informed about their rights.	65.2%	86.4%	NS
the patient and family do not need to be informed about their rights.	34.8%	9.9%	0.001
is hard to say.	10.7%	3.7%	NS
Hospice conditions *			
Should be very good	93.5%	94.0%	NS
At least good	6.5%	0%	0.029
Rooms should be kept clean	99.3%	96.3%	NS
Hard to say	0.7%	6%	NS
Caring for the safety of the sick			
The safety of the sick should be taken care of	99.7%	97.4%	NS
There is no need	0%	1.3%	NS
Hard to say	0.3%	1.3%	NS
The need to ensure intimacy to the patient			
Yes	98.9%	94.0%	NS
Hhere is no need	0%	2.4%	NS
hard to say	1.1%	3.6%	NS
The need to ensure the dignity of the patient			
Yes	98.3%	92.8%	NS
There is no need	0%	3.6%	NS
Hard to say	1.7%	3.6%	NS

* Possibility of multiple-choice answers.

**Table 6 ijerph-20-05334-t006:** The respondents’ opinions on the waiting time for admission to hospice.

	Patients’ Families with		*p* Value
	Oncological Surgery Ward (Group I) N = 108	Hospice (Group II) N = 104	
Less than 8 days	47.5%	67.5%	NS
From 8 to 14 days	31.2%	16.1%	NS
Over 14 days	26.2%	16.3%	NS

**Table 7 ijerph-20-05334-t007:** The respondents’ opinions on the need to inform the patient that he/she is in a hospice as well as about illness.

	Patients’ Families with		*p* Value
	Oncological Surgery Ward (Group I) N = 108	Hospice (Group II) N = 104	
Knowledge about the stay in hospice by the patient			
Dependence on physical, mental, and spiritual conditions	68.6%	17.3%	<0.001
Should always know about it	28.5%	75.0%	<0.001
Hard to say	2.9%	9.6%	NS
Patient’s knowledge of the disease			
The sick person should receive as much information as they wish	64.5%	50.0%	NS
It should be fully communicated, as information is the responsibility of the entire patient care team	12.7%	32.7%	0.016
Should not say anything to the patient	20.8%	24.0%	NS
Hard to say	2%	3.9%	NS

**Table 8 ijerph-20-05334-t008:** The respondents’ opinions about the hospice staff *.

	Patients’ Families with		*p* Value
	Oncological Surgery Ward (Group I) N = 108	Hospice (Group II) N = 104	
Composition of hospice staff			
Every doctor	98.2%	16.3%	<0.001
Doctor specializing in palliative medicine	64.8%	78.8%	NS
Oncologist	54.3%	28.8%	0.027
Every nurse	97.5%	34.6%	<0.001
A nurse with a specialization or course in palliative medicine	87.6%	62.5%	NS
Physiotherapist	85.3%	43.3%	0.009
Psychologist	83.4%	59.6%	NS
Nun	45.3%	17.3%	0.005
There should be unlimited access to…			
families	100%	100%	NS
clergyman	98.8%	93.8%	NS
psychologist	98.1%	59.6%	0.046
physiotherapist	94.6%	43.3%	0.002

* Possibility of multiple-choice answers.

**Table 9 ijerph-20-05334-t009:** Opinion on the participation of the family in the care/treatment of a sick person.

	Patients’ Families with		*p* Value
	Oncological Surgery Ward (Group I) N = 108	Hospice (Group II) N = 104	
Definitely yes	68.3%	91.5%	NS
There is no need	12.5%	2.5%	0.037
Depending on the state of health of the sick person	18.8%	0%	<0.001
Hard to say	0.4%	6.0%	NS

**Table 10 ijerph-20-05334-t010:** The respondents’ opinions on the patient’s feeling of loneliness and the expected support in hospice.

	Patients’ Families with		*p* Value
	Oncological Surgery Ward (Group I) N = 108	Hospice (Group II) N = 104	
A feeling of loneliness			
The sick person does not feel lonely	54.7%	62.4%	NS
The sick person feels lonely	35.1%	22.4%	NS
Hard to say	10.2%	15.2%	NS
Who should receive support in hospice *			
Patient	98.7%	85.6%	NS
Family	95.9%	52.9%	0.02
Staff	87.4%	16.3%	<0.001
Who should provide support *			
Clergyman	76.4%	67.3%	NS
Nurse	52.3%	42.6%	NS
People after the loss of a loved one	65.4%	16.3%	<0.001
Doctor	45.7%	35.6%	NS
Nun	11.7%	19.2%	NS
Cleric	7.9%	11.5%	NS

* Possibility of multiple-choice answers.

**Table 11 ijerph-20-05334-t011:** Respondents’ opinions on the desired religious practices in hospice *.

	Patients’ Families with		*p* Value
	Oncological Surgery Ward (Group I) N = 108	Hospice (Group II) N = 104	
Holy Mass	86.7%	87.9%	NS
Rosary prayer	81.6%	31.7%	0.003
Confession	81.2%	5.8%	<0.001
Sacraments	81.2%	5.8%	<0.001
0 Meditation on the texts of the Holy Scriptures	35.7%	17.3%	0.044
Reading religious books	9.6%	10.6%	NS
Way of the Cross	10.2%	13.5%	NS
Individual prayers	9.6%	5.8%	NS
Talking to the priest	3.9%	5.8%	NS
Hard to say	8.7%	4.8%	NS

* Possibility of multiple-choice answers.

**Table 12 ijerph-20-05334-t012:** The respondents’ opinions on the public focus on palliative care.

	Patients’ Families with		*p* Value
	Oncological Surgery Ward (Group I) N = 108	Hospice (Group II) N = 104	
Does not pay much attention	87.3%	59.7%	NS
Pays a lot of attention	9.6%	31.6%	0.002
Hard to say	3.1%	8.7%	NS

**Table 13 ijerph-20-05334-t013:** Opinions on problems in patient care at the end of life *.

	Patients’ Families with		*p* Value
	Oncological Surgery Ward (Group I) N = 108	Hospice (Group II) N = 104	
Increasing number of patients with chronic diseases	96.7%	23.1%	<0.001
Lack of public awareness of palliative care	92.7%	37.5%	<0.001
The lack of knowledge of relatives about the treatment of the dying sick	89.9%	43.3%	0.001
Lack of skills in nursing a chronically ill and dying patient	88.7%	19.2%	<0.001
Lack of family psychological skills	89.4%	3.8%	<0.001
Occurrence of burnout syndrome of members of the therapeutic team	86.7%	7.7%	<0.001
A small number of inpatient centers and palliative care clinics	79.9%	47.1%	0.049
A small number of suitably qualified staff	74.5%	19.2%	<0.001
Too low financial outlays for hospices	68.9%	50.0%	NS
Lack of psychological skills of the staff	65.4%	3.8%	<0.001
Hard to say	45.8%	4.8%	<0.001

* Possibility of multiple-choice answers.

**Table 14 ijerph-20-05334-t014:** Opinions on how to improve patient care at the end of life *.

	Patients’ Families with		*p* Value
	Oncological Surgery Ward (Group I) N = 108	Hospice (Group II) N = 104	
Increasing the number of medical personnel	87.4%	15.4%	<0.001
Increasing the number of beds	82.4%	0%	<0.001
Increasing financial outlays for hospices	56.9%	0%	<0.001
Improving the qualifications of medical personnel	55.3%	0%	<0.001
Better supply of medical equipment	45.6%	0%	<0.001
Showing more kindness, and support for patients and family	39.2%	92.8%	
Increasing the knowledge of hospice staff regarding communication with the patient and family in care for a terminally ill patient	31.9%	0%	<0.001
Having more conversations with the patient and his family	30.9%	0%	<0.001

* Possibility of multiple-choice answers.

## Data Availability

All data sets on which the conclusions of the paper are based are available upon request to the corresponding author.

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
