# Peer review of "The Functioning of Hospice in the Perception of Family Members of Cancer Surgery and Hospice Patients"

_ijerph, 2023, doi:10.3390/ijerph20075334_

Round 1
Reviewer 1 Report
Congratulations on your effort,
I don't wish to express any negative comment for your work. It is an interesting idea, but of course with not a great merit. I would suggest less detail when it comes to tables and results, because it is tiring for the rider.
Author Response
Thank you very much for your remarks. We tried not to provide more details in the tables, and the description of the results is simple.
Reviewer 2 Report
Title
- There appears to be a dyad that must be evidenced in the title. Two groups are present throughout the text: the families and patients. This needs to be clarified in the title.
Abstract
- Background should direct more to the review question. There is no clear link between the first sentence and the study aim;
- Line 22, what do you mean by "influenza II"? Do you mean Group II;
- Abstract requires more articulation between ideas. Some sentences seem odd when reading;
- Conclusion should address groups I and II.
Introduction
- Please address the fact that the text format (e.g. font type and size);
- Line 37, "habits" is repeated;
- Introduction needs to be stronger and address the main topics in some depth and complexity.
Material and methods
- Please describe the type of study. It is not clear if it is a qualitative or quantitative study;
- From the text, the authors appear to have used open-ended questions. If so, there seems to have been a coding of the questions. Please indicate how this has been performed;
- According to the tables presented, there is a high probability of the Chi-square test assumption not being met ("No more than 20% of the expected counts are less than 5 and all individual expected counts are 1 or greater").
Results
- Some tables' sum surpasses 100,0%. For example, table 4, group II, if summing lines 1 to 4, we have a total of 101.9%. This may be accurate, but since the material and methods are so poor in detail, it is hard to understand what the results mean.
Discussion
- Some findings need to be found with the study's aim. Focus clearly on the functioning of the hospice. As an example, the discussion starts with the opinion related to euthanasia. Why this option? Do the authors relate the functioning with euthanasia?
Conclusions
- The conclusion does not appear to address the findings related to group II.
Overall, the study needs more articulation and detail to allow the readers a better understanding of the importance of the research and replicate it if intended.
Author Response
Title
- There appears to be a dyad that must be evidenced in the title. Two groups are present throughout the text: the families and patients. This needs to be clarified in the title.
We have changed the tititle.
The functioning of the hospice in the perception of family members of cancer surgeryand hospice patients
Abstract
- Background should direct more to the review question. There is no clear link between the first sentence and the study aim; We have changed the Background.
The palliative care service in Poland is for all dying people and their families to have timely access to quality palliative care services
- Line 22, what do you mean by "influenza II"? Do you mean Group II. It must be Group II
- Abstract requires more articulation between ideas. Some sentences seem odd when reading;
- Conclusion should address groups I and II.
We have changed the conclusions.
Most of the families of patients from the oncological surgery departments Group I and the hospice Group II had positive first associations with the hospice. However, families from Group II had more critical remarks on the hospice functioning.
Introduction
- Please address the fact that the text format (e.g., font type and size); We have corrected it.
- Line 37, "habits" is repeated; We have removed it.
- Introduction needs to be stronger and address the main topics in some depth and complexity.
Palliative care and hospice care are designed to relieve pain and distressing symptoms of the disease and to maintain the patient's quality of life at the highest possible level.
In Poland, the National Health Fund finances several types of such care: outpatient – palliative medicine clinic, stationary – hospice or palliative medicine department in a hospital
home – home hospice.
Palliative and hospice care covers mainly people with: cancer, AIDS, consequences of diseases of the central nervous system, some types of respiratory failure, cardiomyopathy,
chronic wounds, ulcers from bedsores.
Material and methods
- Please describe the type of study. It is not clear if it is a qualitative or quantitative study; This is a quantitative study.
- From the text, the authors appear to have used open-ended questions. If so, there seems to have been a coding of the questions. Please indicate how this has been performed; We did not use open-ended questions but the
- According to the tables presented, there is a high probability of the Chi-square test assumption not being met ("No more than 20% of the expected counts are less than 5 and all individual expected counts are 1 or greater").
For counts lesser than 5 the Fishr exact test was used.Materials and methos.
Results
- Some tables' sum surpasses 100,0%. For example, table 4, group II, if summing lines 1 to 4, we have a total of 101.9%. This may be accurate, but since the material and methods are so poor in detail, it is hard to understand what the results mean.
We have correcte d it.
Discussion
- Some findings need to be found with the study's aim. Focus clearly on the functioning of the hospice. As an example, the discussion starts with the opinion related to euthanasia. Why this option? Do the authors relate the functioning with euthanasia?
We have removing the sentences on euthanasia?
Conclusions
- The conclusion does not appear to address the findings related to group II.
We have corrected the conclusions.
Most families from both studied groups had positive first associations with the hospice; they did not fear it when they thought about it and would recommend it to other families as a form of care for the sick.
The diversity of the opinions of the surveyed families on the functions of the hospice were noted. For example, families of hospice patients (Group II) had more remarks on the scope of information, the role and composition of the staff, the participation of families in the care/treatment of the patient, the patient’s feeling of loneliness, the expected support and desired religious practices in the hospice.
Families of patients from oncological surgery departments were more convinced than the families of patients in hospice that society does not pay much attention to palliative care.
Reviewer 3 Report
Authors should be congratulated for their work. The topic is interesting. To date, the management of people suffering from oncological disease represents a hot topic and oncological diseases are now considered socially affecting diseases. However, it is very difficult to read the manuscript: it is redundant and does not help the reader to keep attention. Some parts of the manuscript are still in Polish and there are repetitions of words. Moreover, the quality of the table is very low. Thirteen of the 29 references are Polish, so it is very difficult to check them.
I suggest a major revision in order to address these manuscript gaps.
Author Response
Authors should be congratulated for their work. The topic is interesting. To date, the management of people suffering from oncological disease represents a hot topic and oncological diseases are now considered socially affecting diseases. However, it is very difficult to read the manuscript: it is redundant and does not help the reader to keep attention. Some parts of the manuscript are still in Polish and there are repetitions of words. We have provided some new sentences and we have removed the repetitions of words.
English has been corrected by The EditMyEnglish staff. EditMyEnglish | Professional Proofreading and Editing Services :: EditMyEnglish, we are committed to providing high-quality, accessible editing services to people all over the world. In fact, our staff have edited over 125 million words since 2004 and served over 10,000 customers from 125 countries. We offer high quality, fast turnaround editing services at very competitive prices. You can find English language editing and proofreading for any kind of document – that is, from essays, to homework assignments, to journal manuscripts, to books, to letters, to company newsletters, to white papers, to website text, and much, much more!
Moreover, the quality of the table is very low. In our opinion the tables are simple.
Thirteen of the 29 references are Polish, so it is very difficult to check them.
I suggest a major revision in order to address these manuscript gaps.
We have provided non- Polish references there are now 32 references and only 8 references from Poland.
We enclosed the certificate text correction.

Reviewer 4 Report
The study aimed to assess the perception of the role of hospice by families of oncological patients. The study included 211 family members of cancer patients. Families of patients from oncological surgery departments saw problems in patient care at the end of life. Still, most of the families of patients from both the oncological surgery departments and the hospice had positive first associations with the hospice. The manuscript reviews 29 articles regarding this topic. The topic of this manuscript is up-to-date, attractive and well-suited for your journal. The manuscript is well-written and divided into 6 main parts, the text is clear and easy to read. For better visualisation authors used 14 tables. I suggest checking for some spelling mistakes and grammar errors. Otherwise, I have no major concerns about this manuscript and recommend it for publication.
Author Response
The study aimed to assess the perception of the role of hospice by families of oncological patients. The study included 211 family members of cancer patients. Families of patients from oncological surgery departments saw problems in patient care at the end of life. Still, most of the families of patients from both the oncological surgery departments and the hospice had positive first associations with the hospice. The manuscript reviews 29 articles regarding this topic. The topic of this manuscript is up-to-date, attractive and well-suited for your journal. The manuscript is well-written and divided into 6 main parts, the text is clear and easy to read. For better visualisation authors used 14 tables. I suggest checking for some spelling mistakes and grammar errors. Otherwise, I have no major concerns about this manuscript and recommend it for publication.
Thank you for remarks. The article has been corrected by EditMyEnglish staff . We have enclosed the certificate of article correction.

Round 2
Reviewer 3 Report
Authors should be congratulated for their work. They addressed all my concerns. The manuscript is suitable for publication